# Adolescent Victimization during COVID-19 Lockdowns and Its Influence on Mental Health Problems in Seven Countries: The Mediation Effect of Resilience

**DOI:** 10.3390/ijerph20031958

**Published:** 2023-01-20

**Authors:** Mónica Bravo-Sanzana, Rafael Miranda, Xavier Oriol

**Affiliations:** 1Núcleo Científico-Tecnológico en Ciencias Sociales y Humanidades, Universidad de La Frontera, Temuco 4811230, Chile; 2Department of Psychology, Universidad Continental, Huancayo 12000, Peru; 3Department of Psychology, Universidad de Girona, 17004 Girona, Spain

**Keywords:** adolescents, victimization, mental health, resilience, psycho-social resources and risks, cross-cultural, COVID-19

## Abstract

The objective of this study was to test the differences between the mean scores of victimization, an indicator of depression, stress, and anxiety (DASS), across seven countries (Australia, Chile, India, Indonesia, Mexico, Poland, and the Russian Federation) during the COVID-19 lockdowns. In addition, this study sought to analyze the mediator role of resilience in these relationships in the different countries. To this end, a structural equation model (SEM) was tested and differences across countries were considered through a multigroup analysis. Data for adolescent students from seven countries (*n* = 7241) collected by the Global Research Alliance showed that levels of anxiety, depression, and stress among adolescents were different in the countries assessed; all of them presented values above the mean of the indicator, with Chile and Russia having the highest values. Regarding the prevalence of exposure to violence, the mean across all countries studied was 34%, with the highest prevalence in Russia and India. At the global level, an adequate adjustment was observed in the SEM mediation model considering all countries. However, a mediator effect of resilience was only observed in the relationship between victimization and the indicator of DASS in Chile, Indonesia, and Russia. The results are discussed, analyzing the relevance of resilience as a protective factor for mental health during COVID-19 lockdowns.

## 1. Introduction

The COVID-19 (SARS-CoV-2) pandemic declared by the World Health Organization in March 2020 has had significant psychological and social effects on populations [1]. For example, fear of COVID-19 has been related to high levels of depression, stress, and anxiety in distinct population groups [2,3,4,5]) and sleep problems such as insomnia and nightmares [6,7]. However, adolescents deserve special attention due to the high prevalence of mental health problems already reported among this population before the COVID-19 pandemic [8]. According to the literature, mental disorders in adults start during adolescence [9]. Furthermore, the pandemic context has caused adolescents to experience a wide range of challenging situations in their development stage, increasing their vulnerability to emotional, behavioral, and well-being problems [10,11].

In addition to mental health problems, the pandemic also generated an increase in exposure to different forms of victimization at home [12] or through social networks (e.g., cyberbullying) [13]. Victimization is usually associated with increased externalizing and internalizing problems and total psychological distress [14]. During the pandemic, an increase in mental health problems has been observed in adolescent populations, which makes it crucial to assess the effects of victimization on mental health during the COVID-19 pandemic [15]. This analysis should consider that countries with higher inequality have suffered more regarding violence during the pandemic [16,17].

Our data suggest the need to prevent the effects of the pandemic on adolescent mental health [18]. In addition, different studies reveal that some variables such as resilience act as protecting factors for mental health and victimization [19], which is an incentive to propose guidelines for addressing the psychological consequences of the pandemic. From this perspective, cross-cultural studies on the topic are scarce. Therefore, the objective of this study is to examine the mediating role of resilience in the relationship between victimization and levels of depression, anxiety, and stress in seven countries (Australia, Chile, India, Indonesia, Mexico, Poland, and the Russian Federation) during the COVID-19 lockdowns.

### 1.1. Adolescence and Mental Health Problems during the Pandemic

Adolescence is a vulnerable developmental period that involves important biological and psychosocial changes and specific needs, such as interaction with peers and participation in risk activities [20]. From this perspective, psychological problems are common among adolescents and are strongly influenced by stressful events [21].

In the context of the COVID-19 pandemic, some countries established measures such as lockdowns, social distancing, suspension of school, and implementation of online learning, among others. Therefore, it is no surprise that initial research on the effects of the coronavirus pandemic on adolescents has found a high prevalence of depression symptoms and anxiety in Chinese adolescents [22], that being a woman was the greatest risk for Chinese and Spanish adolescents [21,23], or that fear of the disease was a predictor of acute stress in adolescents from Spain and the Dominican Republic [24,25]. Different systematic reviews also show that adolescents suffered higher levels of anxiety, depression, and stress in countries such as Germany, Canada, China, Denmark, the USA, the Philippines, Japan, the United Kingdom, and Italy during the COVID-19 pandemic [26,27].

Nevertheless, this topic has scarcely been addressed in the countries of this study. For example, research has revealed that in Australia, adolescents experienced a decay in their mental health from the start of the pandemic, with negative effects on learning, friendship, and family relationships [28]. In Chile, family and health problems increased emotional problems among adolescents during the pandemic [29]. In India, a study conducted prior to the COVID-19 pandemic reported that, according to the National Mental Health Survey (2015–2016), the prevalence of psychiatric disorders among adolescents (13–17 years of age) was approximately 7.3%, and this percentage is estimated to have increased exponentially after the pandemic [30,31]. In Indonesia, a study reported that during phases 1 and 2 of the COVID-19 pandemic, which involved the suspension of schools, adolescents were at risk of emotional and behavioral problems [11]. In Mexico, a study showed that approximately 21% of students reported suicidal behavior during the COVID-19 pandemic, the variables most associated with this behavior being “female sex, depression, hopelessness, anxiety, alcohol tobacco consumption and childhood trauma” [32]. In Poland, a study of the relationship between the activities of young people (in both a virtual environment and the real world) and their mental health during the period of social isolation caused by the COVID-19 pandemic found that negative affectivity was intensified in adolescents who spent more time in online environments. On the contrary, engaging in physical and social activities outside the internet acted as a protective factor for the mental health of adolescents in that period [33]. Finally, in Russia, a study assessed the effect of self-isolation during the COVID-19 pandemic on the emotional well-being of different groups of adolescents. The results showed that self-isolation during the period of quarantine caused serious problems in different areas such as academic studies, communication with peers, and contact with family members. Specifically, adolescents hospitalized in a children’s psychiatric clinic were those who presented the greatest difficulties in these different areas during this period [34]. The above indicates that more research that is aimed at detecting the risk and protective variables affecting the mental health of adolescents during the COVID-19 pandemic is necessary [23].

### 1.2. Victimization and Mental Health Problems during Adolescence

As mentioned above, the pandemic has caused unusual situations in all countries due to lockdowns and other protective measures taken against COVID-19 [13]. Consequently, different studies have shown an increase in domestic violence, especially in middle-low-income countries where there is higher vulnerability due to structural violence [35,36]. For example, studies in Chile have revealed that child and adolescent victimization is a major problem in the country, with adolescent victimization percentages above those of Europe and North America [37]. In addition, victimization has been associated with depression symptoms and frequent self-aggression, which suggests that this psychopathology may be related to child victimization experiences [38]. Recent studies in Mexico have indicated that victimization scores for externalizing and internalizing symptoms had more variation than in a single type of victimization, concluding that conventional offenses should be dealt with in a specific way when assessing Mexican adolescents [39] and that child abuse in the family seems to be associated with the risk of peer victimization [40]. Another study focusing on harmful peer aggression in four world regions, including adolescents from Australia, India, Indonesia, Mexico, and Poland, among other countries, found that one-third of peer aggression could be attributed to friends (excluding best friends) and classmates/peers. Likewise, best friends only experience damage when relational aggression such as spreading rumors or being cast aside was involved. There was a trend whereby women reported more damage from peer aggression than men. Despite cultural variations, there were similarities in the level of damage experienced in different types of relationships, with the highest levels corresponding to relational aggression from best friends. Among differences by country, India and Indonesia adolescents reported a greater level of maximum harm when compared to other locations (Australia, Greece, India, Indonesia, Israel, Italy, Mainland, China, Poland, South Korea, Spain, Taiwan and The Philippines) [15]. A study in Poland found that most of the sample (70%) experienced more than one type of victimization in the previous year; in addition, community disorganization, low commitment to school, poor family management, family conflict, peer social preference, and teacher-rated withdrawn and disruptive behavioral problems were identified as predictive factors for victimization [41]. In the same vein, evidence shows that victimization is more frequent in middle-low-income countries than in high- and upper-middle-income countries since adolescents are more prone to experiencing risk factors during the different development stages [42]. Adversity (such as that caused by the current pandemic) can leave adolescents more vulnerable to different forms of victimization [12], which is associated with higher probabilities of developing mental health problems and participating in behaviors that are risky for health [42]. The health problems reported during the pandemic are related to depression, anxiety, low self-esteem, and difficulties in interpersonal relationships, among other factors [43,44]. Some studies have already proven that countries such as China and the USA have experienced an increase in domestic violence during the pandemic [45,46], but data is lacking on other countries, specifically middle-low-income countries. To summarize, recent studies have demonstrated that the number of adolescent victimization cases increased with COVID-19 and that many lockdown measures promoted by countries have also triggered more mental health problems in adolescents. Since previous studies have demonstrated that adolescents who suffer some type of victimization present more risk factors for mental health problems, more studies are necessary to understand whether the increase in adolescent victimization during the pandemic is also related to higher levels of stress, anxiety, and depression.

### 1.3. The Mediation Effect of Resilience

Resilience as a construct refers to the maintenance of positive adaptation by individuals, despite experiencing significant adverse events [47]. In this sense, the literature has numerous definitions and research comprises different levels of analysis [48]. The current definitions of resilience include three characteristics: (a) trait resilience, which states that personal qualities operate to cause people to thrive in adverse conditions, deal with stress, achieve good adjustment, and improve their physical and mental health [49,50]; (b) result-oriented resilience, which is understood as a behavioral problem outcome to overcome adversity [51]; and (c) process-oriented resilience, where resilience is understood as a dynamic process that includes the interaction of personal attributes with environmental circumstances [47]. These definitions have in common the capacity of an individual for adapting and successfully facing adversity in a process that involves individual traits and environmental situations [52]. At the general level, research on resilience at the adolescent stage has received increased attention as resilience acts as a protective factor for mental health [19,53,54,55,56]. For example, resilient people show lower levels of mental ill-being and higher levels of positive mental health over time [57], which has been related to higher levels of adolescent well-being [58]. From this perspective, a recent meta-analysis revealed that interventions for improving the individual, family, and social resilience factors of young people would reduce the risk of psychopathologies in adverse situations [59]. However, these data are in contrast to other meta-analyses conducted with children and adolescents from middle- and low-income countries that highlighted several variations in the relationship between resilience and mental health outcomes across countries; therefore, sociocultural factors are very important to understand these relationships [60].

Since resilience seems to be an important factor in reducing adolescent mental health problems, different studies have suggested that resilience may play an important role as a mediator and moderator in the relationship between victimization and mental health issues such as depression, anxiety, and stress [21,57,61]. In this sense, the literature indicates that although victimization has a negative relationship with resilience, victims may activate resilience mechanisms to avoid more mental health problems [62]. For example, resilience has been recognized as an important mediator between victimization and bullying victimization in childhood depression [61]. In turn, it can also mediate the relationship between victimization through bullying and self-harm in adolescents [63] and the relationship between victimization and self-esteem and self-efficacy [62]. However, despite the above, research on the underlying role of resilience in the relationship between victimization and mental health problems is scarce [63]. Additionally, there are no cross-cultural studies that confirm the behavior of resilience as a mediator mechanism in adolescent samples from different countries.

In this sense, the objective of this study is to explore whether resilience can be a fundamental underlying mechanism in the relationship between this victimization and depression, anxiety, and stress during COVID-19 lockdowns in different countries.

### 1.4. Present Study

Evidence suggests that although resilience as a construct presents conceptual and methodological difficulties, it exhibits substantial potential as a protective factor in difficult or risky situations such as the COVID-19 pandemic [47]. Therefore, we believe that resilience can act as a protective factor in the relationship between victimization and mental health problems such as anxiety, stress, and depression during COVID-19 lockdowns. For example, resilience has been found to be a protective factor of mental health [64] and improves general health [65].

In this sense, we believe that resilience may have acted as a mediator in the relationship between victimization and mental health problems during COVID-19 lockdowns in different countries, especially in countries with the most inequality, where victimization and mental health problems seem to have been exacerbated during the pandemic [16,17].

Therefore, this study uses a structural equation model (SEM) to verify the relationship between victimization and indicators of stress, anxiety, and depression through the mediation effect of resilience. To test to which degree the relationships between these variables in the model differ across countries, an invariance analysis was conducted. In concrete, this study has the following aims:(1)To test the differences between the mean scores of victimizations and indicators of depression, stress, and anxiety (DASS) between seven countries during COVID-19 lockdowns.(2)To test a structural model that considers the relationship between victimization and indicators of depression, stress, and anxiety (DASS) through the mediation effect of resilience in seven countries during COVID-19 lockdowns.

## 2. Materials and Methods

### 2.1. Participants

This work is part of an international study led by the Global Research Alliance in the context of the COVID-19 pandemic and has the objective of analyzing the well-being, mental health, and victimization of adolescents at the global level. The questionnaire was translated into the respective languages of the countries included in the project.

The study was approved by the ethics committees of Universidad de Flinders (Australia) and local universities. Invitations to participate were sent by electronic means, for example, an e-mail including the access link to the platform. The platform contained an informed consent clause for parents or guardians of students and an informed assent clause for students. Data were gathered during the academic year of 2020 from June to September (during lockdowns) in the context of the COVID-19 pandemic. The survey was anonymous and confidential.

Initially, 32 countries were part of the global study. Sample selection in each country was non-randomized and by convenience. For this study, only countries that reported at least 450 cases were selected. The selection of this cutoff point was determined from the Sopers software [66], which evaluates the sample needed for effect estimation in structural equation models based on the anticipated effect size (λ = 0.2), statistical power level (1-β) of 0.8, significance level (α = 0.01), number of latent variables (3), and number of observed variables (39), which requires a number of 410 observations as a minimum to perform these analyses.

Finally, 6423 adolescents from 7 countries were considered in the study. Table 1 presents the distribution of participants by country.

Regarding sex, 42.4% of respondents were women (*n* = 2721), 31.1% were men (*n* = 1996), and 1.3% stated being non-binary (*n* = 72), with 25.4% (*n* = 1634) not indicating any sex. Mean age was 14.31 (DE = 1.90).

### 2.2. Measures

The scales used for the study are presented below. The reliability, factor analysis fit indexes, and standardized factor loadings of scales by country are presented in Table 2.

Victimization during lockdowns. To obtain an indicator of the victimization suffered during COVID-19 lockdowns, the Student Aggression and Victimization Questionnaire was adapted [67]. It includes eight types of victimization (e.g., “another person (s) spread rumors (fake stories) about me”, “I was beaten, kicked or pushed”, or “somebody was mean to me”). The survey was applied during lockdowns in different countries within the sample and, therefore, referred to victimization suffered during that period. All items were assessed through a dichotomous “yes” or “no” answer. The calculation of this variable was conducted using the means of item scores.

The Depression, Anxiety, and Stress Scale-21 Items (DASS-21). Based on the Lovibond and Lovibond scale [68], DASS-21 is a set of three self-report scales designed to measure the emotional states of depression, anxiety, and stress. Each of the three DASS-21 scales contains 7 items that are divided into subscales with similar content. Participants indicate to what extent they have experienced each symptom during the last week of lockdown using a 4-point Likert scale ranging from 0 “Did not apply to me at all” to 3 “Applied to me very much or most of the time”. In the case of depression, items such as “I was unable to become enthusiastic about anything” or “I felt I wasn’t worth much as a person” were considered in the scale. For the anxiety dimension, items such as “I was worried about situations in which I might panic and make a fool of myself” or “I felt I was close to panic” were part of the scale. In the case of the stress dimension, items such as “I found it hard to wind down” or “I found myself getting agitated (nervous)” were measured.

The Connor-Davidson Resilience Scale (CD-RISC). The Connor and Davidson original scale [49], which was modified by Campbell-Sills and Stein [69], describes resilience as a concept in which personal qualities operate to enable people to thrive in adverse conditions and which, when functional, may improve both physical and mental health. This scale is made up of 10 items (e.g., “I can deal with whatever comes my way.”, “I tend to bounce back after illness, injury, or other hardships.”) and is assessed using a 5-point Likert scale that ranges from 1 = “Not true at all” to 5 = “True nearly all the time”.

### 2.3. Statistical Analysis

First, a confirmatory factor analysis (CFA) was conducted using the MPLUS software 8 v. to determine the unidimensionality of the assessed variables and second-order CFA model for DASS. Following Roth’s recommendations [70], SPSS 26 v. was used for selecting missing data and as a deletion method, with only 4% of reported lost data. Regarding the size of the sample, the scientific literature suggests a 10:1 ratio for calculating the psychometric properties of the scale [71]. The ratio among cases and estimated parameters in this study was 35:1, exceeding the proposed initial ration, which reduces the risk probabilities for type 1 errors. Likewise, following Bentler’s recommendations [72], the following fit indexes were considered to test the model: chi-square, Tucker-Lewis Index (TLI), comparative fit index (CFI)—whose recommended values range from 0.90 to 0.95—and root mean square error of approximation (RMSEA) along with standardized root mean square residual (SRMR)—which, according to Kenny et al. [73], should have values between 0.05 and 0.08. Additionally, the loadings factors were required to be higher than 0.30.

The internal consistency of the final model was calculated using the Omega coefficient [74] and Cronbach’s alpha coefficient. Descriptive statistics and correlations were also computed. Additionally, ANOVA analyses were performed to measure the differences among the countries using SPSS 26v.

Finally, a multi-group analysis was conducted to verify the hypothesis of whether the interviewees of different countries displayed significant differences. First, the invariance of the second-order variable of DASS was calculated. For a second-order CFA model, the following sequence of models was tested according to [75]: Model 0 (without invariance), Model 1 (invariant first-order factor loadings), Model 2 (invariant first-order and second-order factor loadings), Model 3 (invariant first-order factor loadings, second-order factor loadings, and item intercepts), Model 4 (invariant first-order factor loadings, second-order factor loadings, item intercepts, and first-order factor intercepts), Model 5 (invariant first-order factor loadings, second-order factor loadings, item intercepts, first-order factor intercepts, and first-order factor disturbances), and Model 6 (invariant first-order factor loadings, second-order factor loadings, item intercepts, first-order factor intercepts, first-order factor disturbances, and item uniqueness).

For the hypothesis model, the WLSMV estimator was used as the victimization factor was measured using dichotomous variables [76]. In order to meaningfully compare statistics across countries, measurement invariance was required. Three steps were analyzed: (a) configural invariance (unconstrained variables); (b) metric invariance (constrained factor loadings); and (c) scalar invariance (constrained factor loadings and intercepts). Metric invariance allows for a meaningful comparison of correlations and regressions, while scalar invariance enables a meaningful comparison of the latent means [77]. Therefore, we tested each multi-group model in three steps. When any constraint was added to a model, a change in the CFI of more than 0.01 was considered unacceptable [78] since a comparative fit index (CFI) difference not larger than 0.01 across models implies that the model fit does not considerably deteriorate.

## 3. Results

### 3.1. Reliability and CFA Analysis

Table 2 shows that the reliability analyses of each study variable by country were adequate for the DASS variables in all countries with values above 0.90. Resilience also presented high values in Cronbach’s alpha indicator and Omega coefficient above 0.80. Finally, the lockdown victimization indicator showed values equal to or higher than 0.7, except in the case of India, which had an indicator equal to 0.50. Regarding the CFA adjustment indicators, Table 2 shows that the CFI and TLI indicators were equal to or higher than 0.90; in turn, the SRMR and RMSEA indicators showed, in most cases, values equal to or lower than 0.08. Finally, the standard factor loadings reported values higher than 0.30.

### 3.2. Descriptive Analysis

Table 3 presents the descriptive results for each variable under study. The ANOVA analysis revealed significant differences between DASS scores across countries (F = 76.60, *p* < 0.001). The Tukey HSD test showed that the Russian Federation reported higher DASS scores than India (diffmean = 23.93, *p* < 0.001), Indonesia (diffmean = 21.19, *p* < 0.001), and Mexico (diffmean = 16.85, *p* < 0.001). In addition, India and Indonesia presented lower DASS scores compared to other countries. For instance, India presents lower DASS scores than Chile (diffmean = −19.97, *p* < 0.001), Australia (diffmean = −16.94, *p* < 0.001), and Poland (diffmean = −15.50, *p* < 0.001), while Indonesia presents lower DASS scores than Chile (diffmean = −17.22, *p* < 0.001), Poland (diffmean = −12.77, *p* < 0.001), and Australia (diffmean = −14.20, *p* < 0.001).

Regarding victimization scores, the global mean was 0.66 (SD = 1.27) and there were significant differences between countries (F = 22.42, *p* < 0.001). The Russian Federation and India were the countries with the highest values. The Russian Federation presented higher values compared to Australia (diffmean = 0.37, *p* < 0.001), Chile (diffmean = 0.70, *p* < 0.001), Indonesia (diffmean = 0.74, *p* < 0.001), Mexico (diffmean = 0.67, *p* < 0.001), and Poland (diffmean = 0.55, *p* < 0.001). In the case of India, polyvictimization scores presented higher values compared to Chile (diffmean = 0.43, *p* < 0.001), Indonesia (diffmean = 0.47, *p* < 0.001), Mexico (diffmean = 0.40, *p* < 0.001), and Poland (diffmean = 0.29, *p* < 0.01).

Finally, the resilience mean variable at the general level was 34.36 (SD = 9.10), and no significant differences were found across countries (F = 3.10, *p* > 0.05).

### 3.3. Second-Order Invariance Analysis of DASS

Regarding the second-order invariance of DASS 21, the examination of the chi-square differences in Table 4 shows that the first four pairwise comparisons of nested models provide evidence of invariant first-order factor loadings (M1-M0), second-order factor loadings (M2-M1), item intercepts (M3-M2), and first-order factor intercepts (M4-M3). However, the chi-square difference for the comparison of (M5-M4) and (M6-M5) was not statistically significant (∆χ2 (6) = 61.329 and ∆χ2 (7) = 11.61, *p* < 0.05), thus indicating that the constraint of invariant item residual variances (item uniqueness) did not hold. This was also supported by the ∆CFI value for this comparison (−0.011), according to the criterion that ∆CFI < −0.01 indicates a lack of invariance [78].

### 3.4. Structural Equation Model Analysis

At the global level, the model was observed to present adequate adjustment indexes (χ2 = 6989.939, gl = 695.2, TLI = 0.92, CFI = 0.92, RMSEA = 0.04). In this model, the victimization indicator influenced resilience (B = −0.23, *p* < 0.001) and resilience influenced DASS21 (B = −0.30, *p* < 0.001). Regarding the relationship between victimization and DASS21, the total effect reported was B = 0.48, *p* < 0.05, and the direct effect of B = 0.41, *p* < 0.05, meaning there was an indirect effect in the relationship between victimization and DASS21 through resilience (B = 0.07, *p* < 0.01). Likewise, the standardized loading factors presented were greater than 0.30. These results are presented in Figure 1.

Table 5 presents regression indicators from the structural equation model by country and the adjustment indicators of each of them. As observed, except for Indonesia, which presented CFI and TLI indicators slightly lower than 0.90, the other countries had adequate adjustment indexes. At the relationship level, it was observed that the negative relationship between victimization and resilience was higher in the case of India (B = −0.45, *p* < 0.001); however, the relationship between resilience and the DASS21 indicator was not significant for this country (B = −0.09, *p* > 0.05). The most important relationship between these two variables was that in Chile (B = −0.40, *p* < 0.001), followed by Australia (B = −0.35, *p* < 0.001). Regarding the total effects reported between victimization and DASS21, these were significant for Russia (B = 0.54, *p* < 0.001), Mexico (B = 0.52, *p* < 0.001), and Poland (B = 0.49, *p* < 0.001). Regarding the indirect effects reported in the relationship between victimization and DASS21 through resilience, these were observed in the cases of Chile, Indonesia, and Russia (B = 0.10, *p* < 0.01).

Regarding the invariance of the SEM Model, the multi-group models by country in Table 6 did not display satisfactory results when loadings and intercepts were constrained, suggesting that correlations and regressions but not mean scores were comparable among these countries, suggesting the existence of different response styles or different meanings awarded to each scale item in different countries.

## 4. Discussion

This study examined the role of resilience and victimization in adolescent mental health during COVID-19. The first objective of the study was to analyze indicators of mental health problems (anxiety, depression, and stress) and the prevalence of victimization during the COVID-19 pandemic in seven countries (Australia, Chile, India, Indonesia, Mexico, Poland, and the Russian Federation). The results revealed significant differences between countries. Notably, adolescents from the Russian Federation and Chile presented higher indicators of anxiety, depression, and stress during lockdowns, as opposed to India and Indonesia, whose values were lower, and Mexico, which scored low for anxiety. Likewise, global studies conducted in the general population have shown that the impact of COVID-19-related stress factors on mental health (for example, PTSD symptoms, insomnia, and dissociation) comprises a wide range of symptoms and is more severe than other stressful events, especially in Latin America [6,7,79]. The above is in agreement with meta-analytical evidence that the worst mental health levels are found in Africa and South Asia, followed by Latin America [80]. Since this is the case of the general population, it is suggested that the impact on the mental health of adolescents may be higher, especially when considering stress factors for this development stage, such as the reduction in social contact caused by physical distance measures—at least in countries where socioeconomic conditions allowed for this, as inequalities across countries have caused these measures to be considered a privilege [81]. In addition, fear of COVID-19 is another important stressor [24,25].

Differences across countries may be due to the global psychosocial burden related to social inequality, among other associated factors [16,17]. For example, the results reveal that adolescents in India and Indonesia presented the lowest levels of stress, anxiety, and depression during lockdowns. These data may a priori seem surprising, but we believe that a possible explanation for them may be that families did not abide by the strict measures imposed by the government as many countries have large, economically active populations that work informally and, therefore, there might have been high mobility despite compulsory lockdowns [82].

Regarding victimization, the results show a prevalence of 34% at the global level of the sample, which reports having suffered some type of aggression during pandemic lockdowns, with significant differences across the countries studied. It is noteworthy that the Russian Federation and India presented the highest prevalence. Regarding the Russian Federation, representative data from the literature showed that 16% of children were victims of bullying in Russian high schools and that victims tended to avoid reporting these situations of aggression [83]. An increase in domestic violence as a result of COVID-19 [84], especially in middle-low-income countries due to significant structural violence [35,36], could explain the prevalence values found in this study. In connection, the new forms of interpersonal relationships that occurred in the family nucleus as a result of lockdowns may have influenced this increase in domestic violence, particularly in more vulnerable families [18]. From this perspective, the fact that countries such as Chile (26.6%) and Mexico (27.2%) presented a lower prevalence of aggression does not imply that these indicators are optimistic; on the contrary, it may indicate that exposure to violence is an alarming reality and suggest that, in places with high rates of adolescent victimization, lockdown measures and social isolation may be an important risk factor for mental health.

The second objective of this study was to test the mediating role of resilience in the relationship between victimization and mental health problems such as depression, stress, and anxiety (DASS). The results of the model comprising all the countries studied showed adequate adjustment levels globally, despite the absence of invariance across countries. Specifically, three of the seven countries showed an indirect effect of resilience on the relationship between victimization and DASS. These results show the relevance of resilience as a protective factor in the relationship between victimization and mental health problems during COVID-19 lockdowns. However, we must be cautious with these results since there are countries where this mediating effect is not observed. Other previous studies demonstrated that in victimization cases such as bullying, resilience may act as a mediator of adolescent mental health [57,61]. However, there are insufficient data to understand whether resilience could also play a mediating role in the relationship between victimization and mental health problems in the specific context of lockdowns during the COVID-19 pandemic, in which the countries within the sample imposed mobility restrictions and adolescents ceased contact with their peers and school attendance [22].

In this sense, these data are especially relevant to understand the importance of resilience as a protective factor under circumstances of global uncertainty. In turn, it should be noted that when comparing results by country, a mediator effect was not observed in all of them. An indirect effect of victimization on DASS through resilience was observed in the cases of Chile, Indonesia, and Russia. Chile and Russia presented the highest DASS scores, while Russia was also one of the countries with the highest prevalence of victimization. This indicates that resilience has acted as a protective factor in the relationship between victimization and DASS in adverse situations with high uncertainty in middle- and low-income countries such as Russia, Chile, and Indonesia. Despite the above, it is noteworthy that India, Mexico, Poland, and Australia had no mediating effect from resilience, despite this variable showing a negative and significant effect on DASS in the case of Mexico and Poland. This also confirms the existence of important cross-cultural differences in terms of the mediator effect of resilience on mental health among countries, as previously shown by other studies [60].

This study has some limitations. First, the analyses were performed with transversal data, which can impede defining the direction of reported relationships. Second, the data reported were self-reported by students and, therefore, future studies should incorporate other relevant stakeholders such as family members.

## 5. Conclusions

The COVID-19 pandemic has led to social isolation, a lack of physical contact, and an increase in domestic violence, among other consequences. In this sense, this study revealed that 34% of adolescents from seven countries have experienced some aggression during the COVID-19 pandemic, which has translated into mental health problems such as depression, anxiety, and stress. These data confirm the importance of promoting preventative actions and programs that contribute to reducing the risk of complex trauma in children and adolescents in different countries due to the impact of COVID-19 [85].

In this sense, our study shows that resilience is a crucial protective factor that should be considered in these actions and programs, which, in turn, should be promoted by public administrations. In concrete, the results indicate that resilience can mediate the relationship between victimization and adolescent mental health problems during pandemic times, for which we believe this is a fundamental factor to understand how to overcome adversity and uncertain situations such as the COVID-19 pandemic, which have consequences at the global level. Although resilience has been demonstrated to be a mediator in the relationship between victimization and mental health problems in different countries, our results also highlight the relevance of considering socioeconomic and cultural factors to understand the possible role of resilience as a protective factor for mental health.

## Figures and Tables

**Figure 1 ijerph-20-01958-f001:**
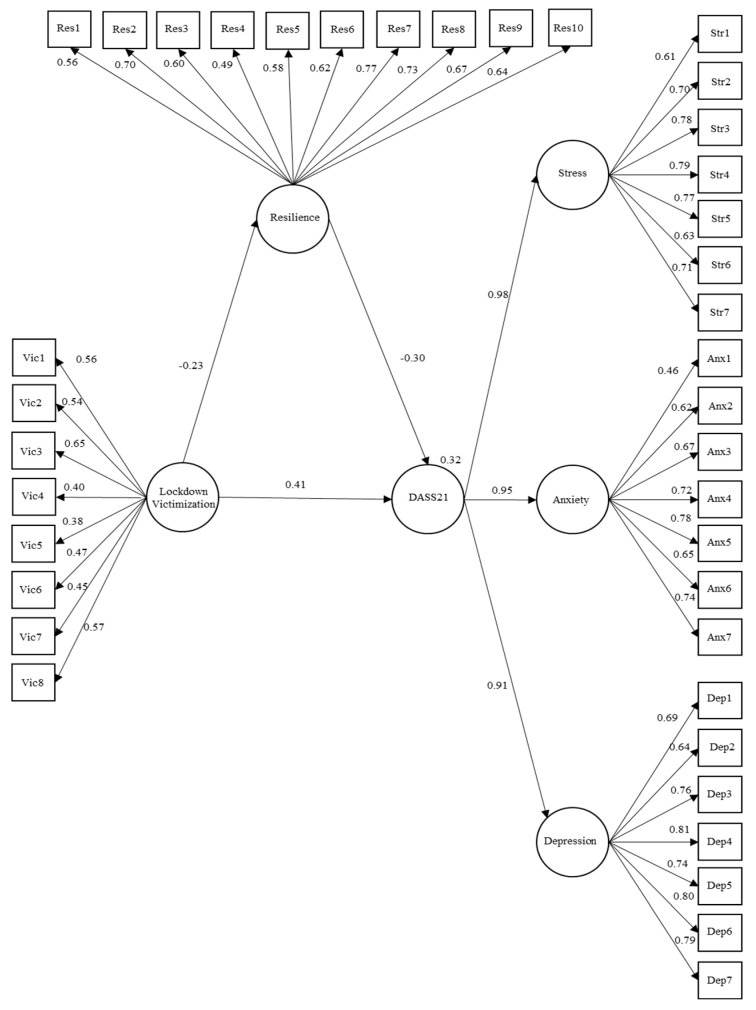
Structural equation global model.

**Table 1 ijerph-20-01958-t001:** Distribution of participants by country.

	Frequency	%
Australia	716	11.1
Chile	2091	32.6
India	795	12.4
Indonesia	722	11.2
Mexico	1186	18.5
Poland	463	7.2
Russian Federation	450	7.0

**Table 2 ijerph-20-01958-t002:** Reliability analysis and confirmatory factor analysis (CFA) fit indexes of the variables.

Country	Variable	α	ω	χ2	CFI	TLI	SRMR	RMSEA	Std Factor Interval
Australia	Lockdown Victimization	0.76	0.78	750.99	0.98	0.97	0.07	0.05	[0.56–0.75]
	DASS-21	0.96	0.96	8808.51	0.92	0.91	0.06	0.08	[0.42–0.97]
	Resilience	0.89	0.89	129.711	0.96	0.95	0.06	0.07	[0.54–0.79]
Chile	Lockdown Victimization	0.72	0.74	754.855	0.94	0.92	0.05	0.03	[0.58–0.90]
	DASS-21	0.95	0.95	1834.442	0.90	0.90	0.06	0.08	[0.48–0.97]
	Resilience	0.89	0.89	273.28	0.97	0.96	0.03	0.06	[0.53–0.76]
India	Lockdown Victimization	0.69	0.71	33.33	0.97	0.96	0.09	0.05	[0.76–0.96]
	DASS-21	0.93	0.94	621.447	0.90	0.91	0.05	0.08	[0.43–0.96]
	Resilience	0.89	0.89	100.79	0.94	0.93	0.07	0.08	[0.61–0.81]
Indonesia	Lockdown Victimization	0.50	0.50	769.89	0.91	0.90	0.10	0.04	[0.50–0.93]
	DASS-21	0.92	0.92	772.795	0.90	0.90	0.05	0.07	[0.39–0.98]
	Resilience	0.83	0.83	221.07	0.91	0.91	0.10	0.09	[0.35–0.75]
Mexico	Lockdown Victimization	0.72	0.74	854.176	0.99	0.98	0.09	0.05	[0.64–0.93]
	DASS-21	0.94	0.94	1139.768	0.91	0.90	0.06	0.08	[0.44–1.00]
	Resilience	0.87	0.87	411.21	0.94	0.92	0.10	0.08	[0.56–0.77]
Poland	Lockdown Victimization	0.71	0.72	870.190	0.98	0.97	0.10	0.04	[0.71–0.88]
	DASS-21	0.95	0.95	605.785	0.91	0.90	0.06	0.08	[0.48–0.95]
	Resilience	0.88	0.89	78.95	0.98	0.97	0.06	0.05	[0.38–0.78]
Russian Federation	Lockdown Victimization	0.70	0.69	838.254	0.98	0.97	0.09	0.05	[0.66–0.92]
	DASS-21	0.93	0.93	610.99	0.90	0.90	0.07	0.09	[0.46–0.96]
	Resilience	0.84	0.84	84.60	0.93	0.91	0.07	0.08	[0.43–0.77]

**Table 3 ijerph-20-01958-t003:** Descriptive analysis of subjective mental health indicators, resilience, and aggression by country.

Variable	Global	Australia	Chile	India	Indonesia	Mexico	Poland	Russia
	M(SD)	M(SD)	M(SD)	M(SD)	M(SD)	M(SD)	M(SD)	M(SD)
DASS	65.57 (27.99)	69.06 (28.72)	72.08 (29.67)	52.11 (24.65)	54.86 (22.57)	59.19 (23.47)	67.62 (26.72)	76.05 (27.26)
Resilience	34.36 (9.10)	34.74 (8.28)	34.52 (9.15)	32.87 (11.44)	34.23 (8.92)	34.71 (8.93)	34.72 (8.66)	33.66 (7.91)
Victimization	0.66 (1.28)	0.85 (1.50)	0.53 (1.16)	0.96 (1.50)	0.49 (0.90)	0.56 (1.21)	0.67 (1.24)	1.22 (1.52)

**Table 4 ijerph-20-01958-t004:** Testing for Factorial Invariance of a Second-Order Factor Model across Countries.

Model	χ2	df	Comparison	∆χ2	∆df	CFI	∆CFI	RMSEA
M0	8744.377	1497				0.926		0.061
M1	8816.551	1515	M1-M0	72.174	18	0.928	0.002	0.061
M2	8842.194	1526	M2-M1	25.643	11	0.925	−0.003	0.068
M3	8918.956	1547	M3-M2	76.762	21	0.917	−0.008	0.073
M4	9003.546	1554	M4-M3	84.59	7	0.914	−0.003	0.074
M5	9064.875	1560	M5-M4	61.329	6	0.903	−0.011	0.080
M6	9076.485	1567	M6-M5	11.61	7	0.893	−0.011	0.082

Note: CFI = comparative fit index; RMSEA = root mean square error of approximation; M0 = baseline model (without invariance); M1 = first-order factor loadings invariant; M2 = first-order and second-order factor loadings invariant; M3 = first-order and second-order factor loadings and item intercepts invariant; M4 = first-order and second-order factor loadings, item intercepts, and first-order factor intercepts invariant; M5 = first-order and second-order factor loadings, indicator intercepts, first-order factor intercepts, and first-order factor disturbances invariant; M6 = first order and second-order factor loadings, indicator intercepts, first-order factor intercepts, first-order factor disturbances, and item residual variances invariant; ∆CFI < −0.01 (signals a lack of invariance).

**Table 5 ijerph-20-01958-t005:** Total and direct effects of victimization on DASS 21 and resilience as a mediator variable.

Variable			Australia	Chile	India	Indonesia	Mexico	Poland	Russia
Victimization	->	Resilience	−0.27 ***	−0.22 ***	−0.45 ***	−0.38 ***	−0.18 ***	−0.29 **	−0.33 ***
Resilience	->	DASS21	−0.35 ***	−0.40 ***	−0.09	−0.28 ***	−0.18 ***	−0.26 ***	−0.32 ***
Victimization ^+^	->	DASS21	0.45 ***	0.44 ***	0.40 ***	0.32 ***	0.52 ***	0.49 ***	0.54 ***
Victimization ^++^	->	DASS21	0.53 ***	0.54 ***	0.44 ***	0.42 ***	0.55 ***	0.56 ***	0.64 ***
CFI			0.93	0.91	0.93	0.89	0.91	0.94	0.9
TLI			0.93	0.91	0.93	0.89	0.91	0.94	0.91
RMSEA			0.03	0.05	0.03	0.04	0.03	0.03	0.04

^+^ Direct effects, ^++^ Total effects, ** *p* < 0.01, *** *p* < 0.001.

**Table 6 ijerph-20-01958-t006:** Multigroup invariance analysis among countries.

Models		χ2	df	*p*-Value	CFI	TLI	RMSEA
Pooled sample		35,576.45	3987	<0.001	0.919	0.916	0.03
Victimization + Resilience Multi-group country groups	Unconstrained	12,623.53	4830	<0.001	0.913	0.914	0.04
Victimization + Resilience Multi-group country groups	Constr. loadings	18,272.75	5268	<0.001	0.905	0.903	0.02
Victimization + Resilience Multi-group country groups	Ctr. load + intercept	18,364.06	5304	<0.001	0.887	0.893	0.02

## Data Availability

Data are available upon request due to privacy/ethical restrictions.

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
