# Peer review of "Adolescent Victimization during COVID-19 Lockdowns and Its Influence on Mental Health Problems in Seven Countries: The Mediation Effect of Resilience"

_ijerph, 2023, doi:10.3390/ijerph20031958_

Round 1

Reviewer 1 Report

Line# 

31-32 - difficult to understand, needs rephrasing - ne of many instances of difficult to follow statements: "Recent studies show the psychological effect of the COVID-19 (SARS-CoV-2) pandemic declared by the World Health Organization in March 2020 (WHO, 2020) on the population."

35  - change "Fear to COVID-19" to "Fear of COVID"

59-62 - "objective of this study" is not clear

78-83 - sentence does not make sense

158-160 - vague summary statement

206 - missing DASS reference?

215-223 - back and forth between pandemic and general resilience without a clear reason for correlations/causation

299/300 - ratio not ration

315 - describe AIC index

317-319 - unclear

Figure-1 - missing description and the following analysis (386-404), including Tables 4 & 5 is not reviewed due to missing method

430 - low indicator meaning? Less DASS?

433 - relevance of culture? Is it a proxy for pandemic lockdown/restrictions? Not clear.

453-455 - unclear objective statement

461-464 - pandemic related factors analyzed in together with non-pandemic factors doe not signify a direct association.

Author Response

Dear reviewer, thank you very much for the suggestions to improve the manuscript. Attached response to comments and suggestions.

Reviewer 2 Report

my reviews are in the attached file

Author Response

(The authors gave the same response as above.)

Reviewer 3 Report

Let me say that I found myself a bit controversial with this manuscript. It started well with a very good introduction, very focused on the relationships among victimization, mental health, and resilience to COVID diseases. I also found the literature review updated and proper as well as the hypotheses and the planned comparison across 7 countries. This latter looked extremely challenging, interesting although risky as well. However, the preconditions of this work were great, and so my mood. Unfortunately, I could not spend the same good words with the methodological part of the article where especially the SEM analyses were not well conducted, I’m sorry to say. Let me start from the participants. Much more details about how the respondents were selected are required. The authors declared that they used a convenience sample, but how it was set up? Any selection criteria? Furthermore, why countries with at least 450 respondents were chosen? Did the author calculate any statistical power to define 450 as a cut-off sampling?

Granted that, the most critic part regarded the measures and the ways how the SEM analyses were conducted especially the measurement invariance issue that constitutes the core of the methodological approach while comparing 7 countries.  In details:

1.       The measures underlining the victimization factor were dichotomous and that’s fine, but the method of estimation for SEM models should have been the Diagonally Weighted Least Square (DWLS) with asymptotic covariance matrix with polychoric (ordinal categories more than 2) and/or tetrachoric (dichotomous) correlations. The method of SEM estimation is never reported in the text.

2.       The CFA results reported in table 2, that should be placed after the Statistical Analysis paragraph, are incomplete. The chi-square, although we know that is often significant for large samples, should have always reported together with the standardized factor loadings. By the way some RMSEA are over 0.10 for some factors and this can be configural problematic, it is possible that some measures are not reliable or valid or still not significant and thus they might be eliminated or modelled by means of residual analysis that is also lacking. Furthermore, a CFA for all factors together is needed other than the single CFAs.

3.       From line 316 to 324, a multi-group analysis was mentioned by the authors concerning the assessing of measurement invariance (MI) issue, but no result was reported regarding the MI in the measurement phase. Assessing the MI across 7 countries requires of a substantial and precise sequence of steps and not only at the structural level with multi-group analysis involving the structural path models as reported in table 5, but firstly at the measurement level with multi-group CFAs. I mean that a first MI is necessary across the measures and thus running multi-CFAs with assessing at least Configural and Metric Invariance to make un-standardized factor loadings comparisons to be meaningful across the 7 countries. If you want to compare the standardized factor loadings also variances should be invariant across the 7 countries. Moreover, the DASS-21 scale is indeed a Second-order factor model loaded by 3 first order factors (i.e., Stress, Anxiety and Depression) a thus also a second-order measurement invariance should have been performed at the outset across the 7 samples. After that, you can run a MI for the structural models with fixing the paths to be equal across the 7 samples together with fixing the factor loadings to be equal (i.e., metric invariance); and thus to make comparisons among un-standardized path coefficients. According to the results depicted in table 5, cross-cultural variance exists and thus it is worthwhile understanding a what level this variance acts and, on the other side, which are the countries where the invariance is satisfied, instead. Working in this way can help the authors to individualize common country-areas of invariance concerning the involved factors. Even though the authors commented such lack of invariance in the discussion, it requires of much more formal analyses in advance. Also running a global structural model looks weak while a complete invariance across all 7 countries is not achieved, and, to say all things, it was very harsh from the beginning to get to such a goal with having so culturally different countries. So, a cultural strategy based on initial potential country similarities, not just cultural, but based on COVID affording plans for instance, should have been reasonably postulated to help these comparisons in resilience mediation.

4.       At last, but not least, although I appreciated the ANOVA analyses, I do think that they are useless here while applying SEM framework, or merely only just descriptive, because the means can be culturally variant also and thus a scalar invariance needs to be assessed as a further MI step for making proper comparisons at the means level.

Conclusively, I do suggest to strongly revise the SEM analyses, especially the MI ones. I do think that the authors will be able to do that, but far more work is needed.

Eventually, I list here some good references to properly run a measurement invariance and means structure analyses along with the way how reporting the results:

Cheung, G. W. (2008). Testing Equivalence in the Structure, Means, and Variances of Higher-Order Constructs With Structural Equation Modeling. Organizational Research Methods, 11(3), 593–613. https://doi.org/10.1177/1094428106298973

Fan W., Hancock G. R. (2012), Robust means modeling : An alternative to hypothesis testing of independent means under variance heterogeneity and non-normality, « Journal of Educational and Behavioral Statistics », 37, pp. 137-156.

Finney S. J., DiStefano C. (2013), Non-normal and categorical data in structural equation modeling, in Structural Equation Modeling : A Second Course, Eds G. R. Hancock, R. O. Mueller (2nd ed.),Greenwich, CT, Information Age Publishing, pp. 439-492.

Leitgob et al. (in press). Measurement invariance in the social sciences: Historical development, methodological challenges, state of the art, and future perspectives. Social Science Research, https://doi.org/10.1016/j.ssresearch.2022.102805

Thompson, M.S., & Green, S.B. (2006). Between-group differences in latent variable means. In G. R. Hancock & R. O. Mueller (Eds.), Structural Equation Modeling: A Second Course (pp. 119-169). Greenwich, CT: Information Age Publishing.

Steenkamp, J.-B. E. M., & Baumgartner, H. (1998). Assessing measurement invariance in cross-national research. Journal of Consumer Research, 25(June), 78–90.

Vandenberg, R.J., & Lance, C.E. (2000). A review and synthesis of the measurement invariance literature: suggestions, practices, and recommendations for organizational research. Organizational Research Methods, 3 (1), 4-69.

Author Response

(The authors gave the same response as above.)

Round 2

Reviewer 3 Report

Let me say that I was very glad to noticing that the authors decided to consider all my remarks and suggestions. They worked hard to improve their manuscript and satisfactorirly well. I was also happy to get that the measument models improved configurally. Also the measurement invariances steps were well-conducted at first and second order level, although M5 and M6 were not necessary, the invariance test could be stopped at the intercepts level. By the way, it was expected that at the means level the invariance was not satisfied because cultural differences exist in so much different countries. For this latter reasons, ANOVA results on means should be taken with caution. However, the global model seemed hold as well as the cross-country comparisons.